# MALDI Mass Spectrometry Imaging Linked with Top-Down Proteomics as a Tool to Study the Non-Small-Cell Lung Cancer Tumor Microenvironment

**DOI:** 10.3390/mps2020044

**Published:** 2019-05-28

**Authors:** Eline Berghmans, Geert Van Raemdonck, Karin Schildermans, Hanny Willems, Kurt Boonen, Evelyne Maes, Inge Mertens, Patrick Pauwels, Geert Baggerman

**Affiliations:** 1Centre for Proteomics, University of Antwerp, Groenenborgerlaan 171, 2020 Antwerpen, Belgium; eline.berghmans@vito.be (E.B.); Geert.vanraemdonck@uantwerpen.be (G.V.R.); karin.schildermans@uantwerpen.be (K.S.); Hanny.willems@vito.be (H.W.); kurt.boonen@vito.be (K.B.); inge.mertens@vito.be (I.M.); 2Health Unit, VITO, Boeretang 200, 2400 Mol, Belgium; 3Food & Bio-Based Products, AgResearch Ltd., Christchurch 8140, New Zealand; Evelyne.maes@agresearch.co.nz; 4Department of Pathology, Antwerp University Hospital, Wilrijkstraat 10, 2650 Edegem, Belgium; patrick.pauwels@uza.be

**Keywords:** NSCLC, MALDI mass spectrometry imaging, tissue preparation, Carnoy’s washing procedure, high resolution mass spectrometry, top-down, endogenous peptides

## Abstract

Advanced non-small-cell lung cancer (NSCLC) is generally linked with a poor prognosis and is one of the leading causes of cancer-related deaths worldwide. Since only a minority of the patients respond well to chemotherapy and/or targeted therapies, immunotherapy might be a valid alternative in the lung cancer treatment field, as immunotherapy attempts to strengthen the body’s own immune response to recognize and eliminate malignant tumor cells. However, positive response patterns to immunotherapy remain unclear. In this study, we demonstrate how immune-related factors could be visualized from single NSCLC tissue sections (Biobank@UZA) while retaining their spatial information by using matrix assisted laser desorption/ionization (MALDI) mass spectrometry imaging (MSI), in order to unravel the molecular profile of NSCLC patients. In this way, different regions in lung cancerous tissues could be discriminated based on the molecular composition. In addition, we linked visualization (MALDI MSI) and identification (based on liquid chromatography higher resolution mass spectrometry) of the molecules of interest for the correct biological interpretation of the observed molecular differences within the area in which these molecules are detected. This is of major importance to fully understand the underlying molecular profile of the NSCLC tumor microenvironment.

## 1. Introduction

For many years, it has been established that failing immunity or suppression of the immune response are among the most important factors in cancer development and progression. Both T and B immune cells can recognize malignant (tumor) cells, but in a developing tumor, their function appears to be suppressed in the tumor-immune microenvironment [1]. Immune evasion is one of the hallmarks of cancer [2], especially driven by immune suppression through cytotoxic T-lymphocyte associated antigen-4 (CTLA-4) or programmed cell death protein 1 (PD-1), both immunomodulatory receptors expressed on immune T cells [3], with an emerging amount of preclinical and clinical data centered around the immune checkpoint PD-1 and its ligand PD-L1 [4]. Its interaction is an inhibitory signaling pathway that, in normal circumstances, downregulates immune T cells to maintain self-tolerance and end ineffective immune responses [5]. However, the expression of PD-L1 has been found on many tumor cells and so, these tumors exploit PD-1-dependent immune suppression [1,6,7]. Several studies suggest that the (re-)activation of an antitumor immune response can result in tumor regression, as well as provide clinical benefit [8]. Immunotherapy is one of the most promising approaches to activate an antitumor immune response, as it provides the blockade of immune checkpoints with monoclonal antibodies [9].

Since only a minority of the non-small-cell lung cancer (NSCLC) patients respond well to radiotherapy, chemotherapy and/or targeted therapies, immunotherapy might be a valid alternative in the lung cancer treatment field [7]. NSCLC is generally linked with a poor prognosis and is one of the leading causes of cancer-related deaths worldwide in both women and men [10]. Immunotherapy has already demonstrated antitumor activity in patients with advanced non-small-cell lung cancer, even when they are previously treated [7]. NSCLC includes three different subtypes: (i) adenocarcinoma, (ii) squamous cell carcinoma and (iii) large cell carcinoma. These subtypes can start from different types of cells in the lung, but the approaches to treatment and prognosis are often similar and the majority of the NSCLC patients are diagnosed with locally advanced or metastatic disease [11].

Pembrolizumab, Nivolumab and Atezolizumab, specific immunotherapy treatments for NSCLC, are approved by the Food and Drug Administration (FDA) to treat patients with advanced and metastatic NSCLC [12,13,14,15]. Both Pembrolizumab and Nivolumab are antagonists of the immune checkpoint PD-1, while Atezolizumab is an antagonist of the PD-L1 ligand, all with the purpose to inhibit the interaction between PD-1 and its ligand PD-L1. These immunotherapy treatments correlated with a significant improvement in overall survival and response rate, in terms of acceptable side-effects and anti-tumor activity, when PD-L1 expression is at least 50% on tumor cells [7]. However, too many patients are selected for immunotherapy treatment, based upon 50% PD-L1 expression criteria, but do not clinically respond to the therapy and demonstrate severe side effects [16]. Evidently, this needs to be avoided as much as possible, for a (cost-) efficient treatment of the NSCLC patient to attain a maximum therapy response and minimized toxicity for a better quality of life [12,17].

The underlying molecular profile of the NSCLC tumor microenvironment is largely unknown. The unraveling of this profile is fundamental to carefully select whether or not immunotherapeutic approaches will be beneficial for NSCLC patients, based on immune and tumor features [12]. Tumor PD-L1 expression alone does not accurately assess the response to immunotherapy treatment and additive diagnostic approaches are fundamental for identifying patients who might have a clinical response to immunotherapy [7]. To this end, matrix assisted laser desorption/ionization (MALDI) mass spectrometry imaging (MSI) is a powerful tool to produce reliable images of the spatial distribution from a broad variety of biomolecules (e.g., peptides, glycans, proteins, lipids, nucleic acids and metabolites) [18,19,20]. MALDI MSI thus provides both spatial and molecular information without a priori knowledge of the present biomolecules. Thereby, it is a multiplexed analysis that allows the screening of hundreds of analytes simultaneously in a single tissue section [21,22,23]. In contrast to immunocytochemistry-based techniques, this unbiased analysis can, in theory, visualize all molecules that can be ionized.

Prior to MALDI-based imaging, possible contaminants in the tissue slice are removed by rinsing the tissue slices with solvents, followed by covering the tissue with a matrix, typically dissolved in an organic solvent, that enhances the extraction of molecules of interest from the tissue. Additionally, the matrix protects the molecules from fragmentation by laser ablation and aids desorption and ionization of the analytes for the further analysis in the mass spectrometer [24]. As a result, a mass spectrum of each spot on the tissue (i.e., every pixel of the ion image) is generated and both the distribution and the relative abundances of the different *m*/*z* values over the entire tissue slice can be analyzed [25]. The biggest advantage of MSI is that spatially resolved mass spectrometric data are produced without destroying the tissue morphology, making a correlation with histological data possible [26]. This is beneficial for this study as tumor features can vary strongly between consecutive tissue sections and in this way, both histological and molecular information can be derived from one single tissue section.

MALDI MSI has already been used in many clinical research projects, ranging from biomarker discovery to cancer diagnostics [18,26,27,28]. MSI data of lung (cancerous) tissues is currently limited to drug distribution [29,30], lipidomic profiling [31] or proteomic profiling (on FFPE tissues) [32]. In this study, MALDI MSI was employed to (partially) elucidate the underlying molecular profile of NSCLC patients, based on endogenous peptide and intact small protein profiles. In this way, immune-related factors (cytokines, chemokines, growth factors, etc.) can be directly derived from NSCLC tissues to provide crucial insights into the interplay and communication between tumor cells and adjacent immune cells. Considering the fact that tumors can be very heterogenous, mass spectrometry imaging has the advantage that it can provide information on both the identity and the localization of these compounds in relation to the localization of cells and structures—information that is lost when only working with tissue extracts in classical peptidomics/proteomics analyses.

Fresh frozen tissue sample preparation, prior to matrix deposition, is regarded as the most critical part of the MALDI MSI workflow. In addition, it is tissue dependent and it is thus a challenge to optimize lung tissue preparation in order to obtain good quality MSI images. Chemical treatment of tissue sections involves removing interferences (e.g., biological salts and lipids, which strongly reduces ionization efficiency of endogenous peptides and small proteins), without causing loss and delocalization of analytes of interest. In this study, twelve chemical treatment steps, which were previously described in the literature for imaging of other types of tissues, are examined to evaluate their potential for peptide imaging in lung cancer tissues [21,33,34,35,36,37,38]. For these purposes, we used human lung cancerous with adjacent reference tissues (lung periphere tissues), to study possible delocalization and to visualize the interaction border between the tumor region and the reference region. Hereby, matrix deposition should also be taken into consideration since a compromise between spatial resolution (limiting diffusion of analytes) and spectral quality (sufficient extraction from the tissue) has to be found. Very thin matrix layers of very small crystals should be deposited, without causing delocalization of the analytes. To reduce the variance of environmental conditions and achieve a homogenous matrix, an automated sprayer is the most suitable device for matrix application for the purposes of this study to maintain reproducibility [28,35].

As peptide and protein identification and quantitation with MALDI MSI is cumbersome and often not possible, liquid chromatography higher mass resolution mass spectrometry-based technologies need to be implemented for reliable identification of the detected peptides with an interesting distribution throughout the lung cancer periphere tissue. This is required for a correct biological interpretation of the detected molecules within the area in which they are detected [18,39]. The purpose of this study is to directly link the masses of the identified peptide(s) from whole lung tissue extracts with the observed mass(es) in mass spectrometry images. For this reason, classical proteomic approaches with enzymatic digestion cannot be performed, as the digested mixture of short peptides complicates the direct linkage of *m*/*z* values of intact molecules observed with MSI. Therefore, top-down peptidomics/proteomics will be used, where intact molecules without enzymatic digestion are analyzed in the mass spectrometer and fragmented using suitable fragmentation methods. This has the advantage that information about possible post-translational modifications (PTMs) is retained and the obtained *m*/*z* value of a single molecule corresponds the *m*/*z* value of the intact molecule observed in mass spectrometry images, after correcting these *m*/*z* values for multiple charges. In addition, the inherent low mass accuracy of MALDI MSI, caused by the uneven tissue thickness surface, and the limitation of identification can be complemented by the higher mass accuracy approaches for a reliable identification of the observed MSI targets [40].

## 2. Materials and Methods

### 2.1. Materials

Acetonitrile, methanol, isopropanol and water (LC-MS graded) were purchased from Biosolve (Valkenswaard, The Netherlands). Ethanol, chloroform, 2,5-dihydroxybenzoic acid (DHB) and ammonium acetate were purchased from Merck (Overijse, Belgium). Xylene, formic acid and trifluoroacetic acid were purchased from Thermo Fisher Scientific (Merelbeke, Belgium). Hexane was purchased from Thermo Fisher Acros Organics (Geel, Belgium). The hematoxylin and eosin staining kit, Quick-D mounting medium and formaldehyde were purchased from Klinipath (Olen, Belgium) and glacial acetic acid and acetone were purchased from VWR Chemicals (Leuven, Belgium).

The human biological material used in this publication was provided by Biobank@UZA (Antwerp, Belgium; ID: BE71030031000); Belgian Virtual Tumorbank funded by the National Cancer Plan [41]. The presented data were derived from two lung cancer patients diagnosed as squamous cell carcinoma and from one patient diagnosed as adenocarcinoma.

### 2.2. Tissue Sectioning and Preparation

Fresh frozen human lung cancerous tissues were collected and tissue sectioning was performed on a LEICA CM1950UV cryostat to obtain slices with 14 µm thickness. To avoid optimal cutting temperature (OCT) compounds, the tissue was attached to the cryostat holder with a water droplet. Tissue sections were thaw-mounted on Indium Tin Oxide (ITO)-coated glass slides (Bruker Daltonik GmBH, Bremen, Germany) and stored at −80 °C prior to use.

For endogenous peptide analysis, a fresh frozen human lung cancer tissue diagnosed as squamous cell carcinoma (one of the three NSCLC subtypes) was serially sectioned at 14 µm thickness. For the analyses, two tissue sections were collected per slide for measurements in duplicate. Tissue sections were desiccated in vacuum for 1 h, followed by a chemical rinsing procedure in a Petri dish while gently swirling. Different used washing procedures are listed in Table 1.

After every rinsing procedure, glass slides were tilted to maximize solvent removal. After the last rinsing part, a vacuum desiccation step of 30 min was performed followed by matrix deposition of 12 layers of DHB matrix (40 mg/mL in 60/0.1 (v/v) acetonitrile/trifluoroacetic acid) by using a SunCollect pneumatic sprayer (SunChrom, Friedrichsdorf, Germany). Matrix layers were applied with an air pressure of 2 bar and with a flow rate of 10 µL/min for the first layer, followed by 20 µL/min for the second layer, 30 µL/min for the third and from then on 50 µL/min, without any drying time between the matrix layers. Lastly, 0.5 µL of peptide calibration standard (Bruker Daltonik GmBH, Bremen, Germany) was spotted on top of the matrix layers, next to the tissue section for external calibration.

### 2.3. Mass Spectrometry Imaging

MALDI MSI data were acquired with a rapifleX tissuetyper in single TOF mode (Bruker Daltonik GmBH, Bremen, Germany), equipped with a SmartBeam 3D laser. Mass spectra were the sum of 1000 individual laser shots, with a 90% laser intensity. Mass spectral peptidomic (*m*/*z* range 800 Da–5 kDa) images were obtained in positive reflector mode with a reflector voltage of 3005 V, a sample rate of 0.63 GS/s, a laser resolution of 50 µm and a raster width of 50 µm × 50 µm.

All the spectra are preprocessed with a Top Hat baseline algorithm for baseline subtraction and the resulting overall average spectrum of the ion image is TIC normalized in flexImaging 5.0 (Bruker Daltonik GmBH, Bremen, Germany) after recalibration in flexAnalysis 4.0 with external calibration standard. The results will be further processed in SCiLS lab 2016b (Bruker Daltonik GmBH, Bremen, Germany) and R software (Cardinal) [42].

### 2.4. H&E Staining

After every MSI experiment, each tissue section was hematoxylin and eosin (H&E) stained according to conventional protocols [43]. The matrix was removed with 70% EtOH, after which the tissue was dried using a vacuum desiccator. The tissue section was then AFA (combination of 70% ethanol, 5% formalin and 5% acetic acid) fixated, followed by a hematoxylin staining for 5 min and the eosin staining was performed for 30 s. The tissue section was rinsed in graded ethanol series (70%, 2 × 95%, 100%) and in 100% xylene for another 30 s. Coverslips were mounted with Quick-D mounting medium. The tissue sections were in this way re-evaluated by the pathologist of University Hospital of Antwerp (UZA) for confirmation of the observed regions.

### 2.5. Peptide and Intact Small Protein Identification

Identification of relevant mass spectral peaks is necessary to gain functional insight into immunological processes. In order to identify the endogenous peptides observed with MSI, up to 10 lung tissue slices from the same piece of tissue were, as soon as possible after sectioning, collected in peptide extraction solvent consisting of methanol:water:formic acid (90:9:1 v:v:v) and shaked on ice for 30 min. The sample was kept on ice during the whole extraction procedure. First, the peptide sample was sonicated with a bar sonicator twice for 15 s each (Branson Sonifier SLPe cell disruptor). After 15 min centrifugation at 14,000 rcf by 4 °C, the supernatant was collected and methanol could be evaporated by using a vacuum centrifuge concentrator (Savant SPD1010 SpeedVac Concentrator, Thermo Scientific). The lipids were removed by re-extraction with n-hexane. From the remaining aqueous fraction, the peptides were concentrated using an ultra-0.5 mL 10 K centrifugal filter device (Merck) and desalting was performed by solid phase extraction with a Pierce C18 Spin Column (Thermo Scientific) according to the manufacturer’s procedure. The eluted sample was again dried using the vacuum centrifuge concentrator and the sample pellets were stored at −20 °C prior to LC-MS/MS analysis.

### 2.6. LC-MS/MS Analysis

The dried fraction(s) containing the peptides (intact, without enzymatic digestion) were dissolved in 15 μL mobile phase A (2% acetonitrile in HPLC-grade water, 0.1% of formic acid) before separation by reversed phase C18 (RP-C18) liquid chromatography on a nanoAcquity UPLC system (Waters, Milford, MA) using an Acclaim PepMap trap column (3 µm particle size; 100 Å pore size; 75 µm × 20 mm, Thermo Scientific) connected to an Acclaim PepMap RSCL C18 analytical column (2 µm particle size; 100 Å pore size; 50 µm × 15 cm, Thermo Scientific). A linear gradient of mobile phase B (0.1% formic acid in 98% acetonitrile and 2% water) started for 5 min with 5% mobile phase B, followed by a steep increase from 5% to 55% mobile phase B in 110 min, followed by 55% to 100% mobile phase B in 3 min and lasted extra 5 min, followed by a decrease from 100% to 5% mobile phase B in another 2 min and lasted for an extra 5 min, with a flow rate of 400 nL/min.

Liquid chromatography separation was for all samples followed by tandem MS (LC-MS/MS) and was performed on a Q-Exactive Plus mass spectrometer equipped with a nanospray Flex ion source (Thermo Fisher, Waltham, MA, USA). The high resolution mass spectrometer was set up in data-dependent acquisition mode where a full MS1 scan (mass range of 350–1850 *m*/*z*) with a resolution of 70,000 was followed by a high energy collision activated dissociation (HCD) MS2 scan (mass range of 100–2000 *m*/*z*) at a resolution of 17,500. Peptide ions were selected for further interrogation by tandem MS as the twenty most intense peaks of a full scan mass spectrum. The normalized collision energy used was set at 30% in HCD. A dynamic exclusion list of 20 s was applied.

Peptide identifications were generated using PEAKS Studio software (version 7.0, Bioinformatics Solutions Inc., Waterloo, ON, Canada) with a 1% false discovery rate. Parent mass error tolerance was set at 10 ppm and precursor mass error tolerance at 0.02 Da. Possible post-translational modifications (PTMs) were set at a maximum of 3 variable PTMs per peptide and included oxidation (at methionine residue (+15.995 Da)), acetylation (+42.011 Da), amidation (−0.984 Da), phosphorylation (at serine, threonine or tyrosine residue (+79.966 Da)), pyroglutamic acid (pyro-glu) formed from glutamic acid (−18.011 Da) and pyro-glu formed from glutamine (−17.027).

## 3. Results

### 3.1. Comparison of Different Chemical Treatment Steps of Lung Cancer Tissues

The first part of the study included the comparison of the different tissue preparation steps, in order to obtain good quality mass spectral images based on the endogenous peptide profile. The comparison can be divided in the evaluation of two different mass spectral profiles, i.e., the lipid and endogenous peptide profile. For both profiles, evaluation was based on the number of the observed peaks, intensity of these peaks and signal-to-noise. Experiments were performed on serial sections from the same fresh frozen human lung tissue to minimize variability from the sample itself and to ensure that the same molecules were present in each tissue section to allow comparison.

For the evaluation of the lipid profiles (Figure 1), lipids need to be removed as much as possible as this allows the detection of low mass peptides that otherwise are masked by the signal of lipids. We aimed to remove lipids as much as possible since these can strongly reduce the ionization efficiency of the peptides of interest in this study due to ion suppression effects. From Figure 1, we can conclude that lipids are naturally abundant in human lung tissues (No wash (control)), but can be partially removed by rinsing the tissue in chloroform, xylene, acetone and hexane and more completely by Carnoy’s washing procedure.

The evaluation of the endogenous peptide profile included the amount of detected peptides (see Table 2) and peptide extraction efficiency. This last part was limited to the *m*/*z* range 3300–3500 in order to compare the extraction efficiency of three intense peptide signals *m*/*z* 3369.5, *m*/*z* 3440.6 and *m*/*z* 3484.6 that occurred in all samples. As shown in Figure 2, these three peptides were best visible using Carnoy’s washing procedure and acetic acid treatment. Of these two procedures, both most peptide signals (Table 2) as highest signal intensity of the peptide peaks were observed using Carnoy’s washing procedure. Although the acetic acid procedure did perform well in terms of peptide detection, it was clearly outperformed by Carnoy’s washing procedure when it comes to removal of the lipids (Figure 1). These solvent wash procedures are probably too harsh and wash out the peptides or the removal of the lipids is not efficient enough, which can lead to ion suppression effects due to high ion intensity of the lipids.

Considering both evaluations, it is apparent that Carnoy’s washing procedure leads to the most complete lipid removal, thereby yielding the best endogenous peptide signal intensity. Therefore, Carnoy’s washing procedure was selected for sample preparation for imaging of lung cancer tissues.

### 3.2. Evaluation of Peptide Delocalization: Cancerous from Noncancerous Lung Tissue

To study possible delocalization of the endogenous peptides caused by chemical treatment steps of the lung cancer tissue, we used human lung cancerous with adjacent reference tissues (lung periphere tissues), chemically treated using Carnoy’s procedure. In order to distinguish the tumor region from the nontumor tissue region in lung cancer tissues based on the endogenous peptide profile, the imaged tissue section was H&E stained after performing the MALDI MSI experiment. The resulting histological data gives information about the tumor and nontumor region, which is indicated in the figures. From two different NSCLC diagnoses—Figure 3 is an example that has been diagnosed as lung periphere squamous cell carcinoma and Figure 4 is an example of lung periphere tissue with diagnosis of adenocarcinoma—the tumor region could be distinguished from the nontumor region based on the endogenous peptide profile. For each tissue, the overall average mass spectrum, generated in R software (Cardinal), was shown with the indicated selected peptide peaks. Peak picking was performed manually in FlexImaging 5.0 (Bruker Daltonik GmBH, Bremen, Germany) by evaluating the distribution of all individual *m*/*z* peaks. We then assessed whether an individual ion was detected only in the tumor region, only in the nontumor region or whether it shows a differentially expression pattern along both regions, as relative abundances are considered with MSI. The distribution of the selected individual *m*/*z* values is presented in Appendix A.

From these results, it can be concluded that Carnoy’s washing procedure does not cause delocalization of the analytes.

As an example showing that, based on the endogenous peptide profile, the tumor region can be distinguished from the nontumor region, a top-down spatial segmentation test was performed in SCiLS lab 2016b (Bruker Daltonik GmBH, Bremen, Germany) on the lung periphere tissue displayed in Figure 4. During a spatial segmentation test, similar spectra are grouped in one cluster. Bisecting the k means algorithm, a top-down and unsupervised segmentation test was performed and divided the tissue into two clusters (Figure 5) that corresponds with the observed tumor and nontumor regions in the H&E staining, evaluated by the pathologist. The main difference detected in this tissue is the difference between tumor and nontumor region. In addition, based on the very comparable delineation of the different regions between the images obtained with classical staining and MSI, we can conclude that Carnoy’s washing procedure does not cause major delocalization of the endogenous peptides.

Information about the tumor and nontumor region could not be obtained from the MALDI MSI experiments on untreated (no wash) lung tissue samples (data not shown). Chemical preparation steps of lung tissues are thus required to obtain good mass spectral images. All experiments were performed using Carnoy’s washing procedure. Up to ten different lung periphere tissues (four with diagnosis adenocarcinoma and six diagnosed as squamous cell carcinoma) were measured (not all data shown) and in not a single case could obvious delocalization be detected and in all cases the tumor region could be distinguished from the nontumor region based on the individual peptide profile.

In order to obtain good spectral images with a good spatial resolution, different types of matrices, solvents, layers, height of the SunCollect sprayer (SunChrom, Friedrichsdorf, Germany) and speed were tested. Results were not shown, as the use of other matrices besides DHB did not result in good spectral images, nor did changing the medium speed of the sprayer. Further, neither the used solvents nor varying the height of the sprayer affected any aspects of the lung images. Finally, 12 layers of DHB matrix (40 mg/mL in 60/0.1 (v/v) acetonitrile/trifluoroacetic acid), deposited with a medium speed (900 mm/min) and a height of 35 mm of the sprayer, proved to be the perfect condition for a sufficient extraction and good signal intensity of the analytes, without visible delocalization. These conditions were applied in all results shown.

### 3.3. Identification of Discriminative Peptides by Top-Down Peptidomics

To find discriminating *m*/*z* values within lung periphere tissues, tumor and nontumor region were determined by H&E staining. Ions visible with MSI that can distinguish between these different regions can either be picked manually based upon their individual distribution pattern throughout these regions or an automated detection approach can be used performing a ‘Receiver Operating Characteristic (ROC)’ analysis (SCiLS lab 2016b (Bruker Daltonik GmBH, Bremen, Germany)) [44]. ROC analysis was performed on both lung periphere tissues in order to find single *m*/*z* values that could perform as a binary classifier between lung tumor and nontumor classes. ROC analysis was performed with a discrimination threshold value of 0.8 to obtain a peak list of discriminative *m/z* values. An example is shown in Figure 6, where two *m/z* values (*m*/*z* 4961.6 and *m*/*z* 4934.9) are determined as characteristic peptides expressed in the lung tumor region, in both squamous and adenocarcinoma NSCLC tissues, with their corresponding ‘area under the ROC curve (AUC)’ scores, which represent the accuracy of the test; the closer to 1, the more perfect. A full list of all characteristic *m*/*z* values for the lung tumor region in human adenocarcinoma lung periphere tissue is presented in Table A1 (Appendix A). For the lung periphere tissue with human squamous cell carcinoma diagnosis, only *m*/*z* value 4934.9 was found to be discriminative between tumor and nontumor region. Considering their interesting tissue distribution, these MSI targets need to be identified.

Though with current MALDI TOF/TOF instruments, it is in theory possible to obtain fragmentation spectra of peptides detected with MSI straight from the tissue [45], identification directly from the tissue has limitations. Often, identification is difficult because of background, generation of chimeric spectra and issues with mass accuracy [39]. To overcome the current limitation of identification of observed peaks straight from MALDI MSI experiments, peptides need to be extracted from whole NSCLC tissue slices, followed by LC-MS/MS analysis [18,39]. The obtained *m*/*z* values in MALDI MSI were compared to those obtained from the LC-MS/MS analysis, after correcting these latter *m*/*z* values for multiple charges. Obtained fragmentation spectra were analyzed by PEAKS software and searched against the uniprot reference proteome database to obtain peptide identity. In some cases, when the quality of the spectra was not sufficient or the correct parent was not selected for fragmentation, the relevant *m*/*z* values were included in an inclusion list and the sample was re-analyzed, so that whenever one of these MSI targets was detected, these would be selected for fragmentation with priority over other possible precursors regardless of their signal intensity. Figure 7, Figure 8 and Figure 9 show three peptides with different distribution in the tumor and nontumor region and their corresponding Q-Exactive Plus top-down mass spectrum obtained with higher-energy collisional dissociation (HCD) in the LC MS/MS set-up. The fragmentation spectra were deconvoluted and de novo sequenced by using PEAKS software with possible variable PTMs of oxidation, phosphorylation, acetylation, amidation and pyro-glu formed from glutamic acid and from glutamine (with maximum 3 variable PTMs per peptide). The full list of matched b and y ions (deconvoluted) for the identified peptides is reported in Appendix B. The displayed data were acquired using human adenocarcinoma lung periphere tissue. The same peptides were identified using human squamous cell carcinoma lung periphere tissues (data not shown). In both cases, the mass accuracy window was set at *m*/*z* 1.6 to accommodate for the inherent low mass accuracy in MALDI MSI. This inherent low mass accuracy of MALDI MSI is caused by the uneven tissue thickness surface and thereby, *m*/*z* values obtained with MSI and higher resolution mass spectrometric approaches can be slightly deviated [40], as mass accuracy obtained with the rapifleX tissuetyper (Bruker Daltonik GmBH, Bremen, Germany) is merely 50 ppm with external calibration in positive reflector mode, but accuracy and resolution is sufficient to obtain monoisotopic masses of the peptides in reflector mode.

As shown in Figure 7, there is a difference in the expression pattern of thymosin β4 between different diagnosed lung periphere tissues. Thymosin β4 has a higher intensity in the tumor region in the lung tissue diagnosed with adenocarcinoma (upper figure, see also Figure 4), while the situation is exactly opposite in the nontumor region in the lung tissue diagnosed with squamous cell carcinoma (lower figure, see also Figure 3), where the peptide signal is higher in the nontumor region. Overexpression of thymosin β4 in the NSCLC region has been associated with carcinogenesis and tumor progression [46,47], where it is believed to activate cell migration and angiogenesis [47]. This could possibly explain the observed expression profile in the different regions. A link between thymosin β10 levels and NSCLC survival rates has been documented. Higher expression of thymosin β10 (Figure 8) is correlated with lower survival rate in comparison with lower levels of thymosin β10 expression in NSCLC patients [48]. Both thymosin β10 and thymosin β4 were expressed in all ten different NSCLC tissues. Finally, a number of studies suggest that persistent elevation of fibrinopeptide A (of which the distribution and identification is shown in Figure 9) has been correlated with treatment failure and poor prognosis for NSCLC patients [49], as higher expression of fibrinopeptide A is associated with increased thrombin activity that led to a lack of (chemo)therapy response [50]. From all the ten measured NSCLC patient tissues, fibrinopeptide A was expressed in three different patients with a normal intensity and in two patient samples with a low intensity. In the remaining five NSCLC samples, no fibrinopeptide A could be detected.

In summary, for a number of the peptides that displayed a differential expression in the tumor and nontumor region of the lung tissue, the literature provides an appealing biological explanation for the observed distributions. The obtained results illustrate that linking the imaged peptide and the corresponding identification with top-down peptidomics can lead to unique insights into the distribution of molecules in the tissue and help in the biological interpretation of MSI data. This is of major importance to fully understand the underlying molecular profile of the NSCLC tumor microenvironment. Using this approach, potentially new immune response patterns to immunotherapy can be elucidated and new potential targets for cancer treatment can be discovered. Additionally, these preliminary results illustrate very well that combining mass spectrometry imaging and top-down peptidomics can be used for future biomarker analyses.

## 4. Discussion

To study the molecular tumor microenvironment of lung cancer, mass spectrometry imaging (MSI) has been recognized as a powerful tool. In this study, we applied MALDI MSI to profile peptidomic differences in non-small-cell lung cancer tissues, after taking the appropriate sample preparation into consideration. Since, to our knowledge, there is no MSI data on endogenous peptides in human lung tissue, the primary aim of this study was to evaluate different sample preparation protocols for applicability to peptide imaging in lung tissues. The Carnoy’s treatment step has been shown to avoid ion suppression of peptides by lipids, thereby enhancing peptide extraction, without causing delocalization of the analytes of interest. The good performance of Carnoy’s washing procedure is probably due to the different steps used. First, salts and debris are removed with the 70% ethanol rinsing step. Rinsing with 100% ethanol leads to dehydration and fixation of the lung tissue. This avoids delocalization of the analytes and improves the tissue’s stability over time. The actual Carnoy’s fluid removes lipids by rinsing with chloroform. In this way, the phospholipids, which comprise most of the cell membrane, were removed and endogenous (mostly secretory) peptides could be displayed to the tissue surface. The presence of acetic acid in this washing fluid probably leads to an efficient peptide extraction, which is also presented in Figure 2 [21]. This allowed us to distinguish the cancerous lung tissue from adjacent normal lung tissue, based on the distribution of peptides.

The main advantage of MALDI MSI over traditional proteomics is the spatial information of the peptides throughout the tissue, but the identification of these peptides straight from the tissue still remains difficult. However, by linking MSI data directly with top-down peptidomics as performed in this study, it is possible to reliably identify MSI targets with an interesting distribution observed with MALDI MSI. The inherent limited mass accuracy of the MALDI-TOF instrument used in a MALDI imaging setup, which has to do with the heterogeneity in composition and thickness of the tissue slices, can make it difficult to unequivocally link observed masses in MSI with identification from the top-down LC-MS/MS analysis. In tissues with large numbers of endogenous peptides (such as brain, pancreas, etc.), this can be an issue requiring the use of high resolution and high mass accuracy instruments such as FTICR to link the right molecular image to the right molecular identification [39,51]. However, although providing better spectral resolution and mass accuracy using Fourier transform instruments for imaging of intact peptides, FTICR has limitations of its own. Orbitrap instruments typically have a high mass limit of 4 kDa and although recent FTICR instruments can be used for imaging of proteins up to 12 kDa, their sensitivity is limited and the low speed of acquisition hampers the generation of high resolution images. In these cases, combining high resolution images obtained with high speed MALDI-TOF instruments with high mass accuracy date from the FT instrument on the same tissue sample can provide an outcome [52]. In non-endocrine tissues, such as lung with less complex endogenous peptide composition, the mass accuracy obtained with a MALDI-TOF instrument will suffice to link peptide masses from MSI to peptide identification.

Our imaging and identification data demonstrate that MALDI MSI could accurately identify differential features in lung cancerous tissues. Our preliminary results show that linking peptide identifications (obtained via top-down approaches) with the distribution of these peptides in the tumor provides biologically relevant data. This approach thus opens opportunities to study the role of peptides in many tissue context-related questions such as tumor heterogeneity, where there is a need to elucidate the role of peptides, growth factors, cytokines, etc., in the interplay between tumor cells and adjacent immune cells within the tumor microenvironment. Though it is worthwhile to note that with our current setup the maximum size of the peptides that can be observed is still limited, we strongly believe in the potential of the combination of MSI and top-down approaches to obtain trustworthy identification in a tissue location context.

## Figures and Tables

**Figure 1 mps-02-00044-f001:**
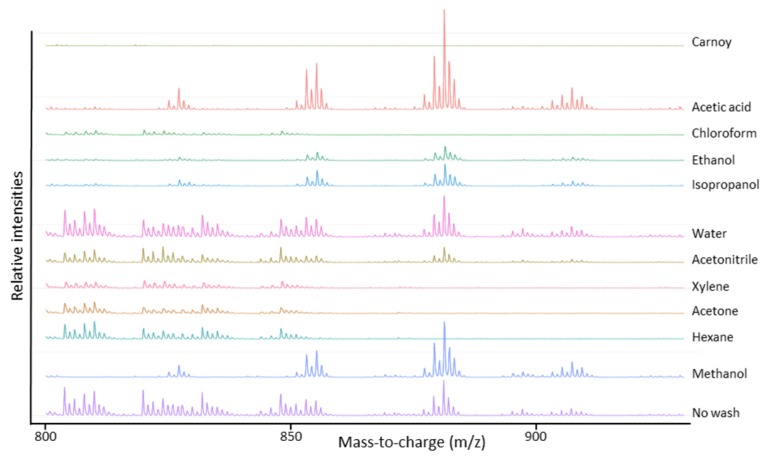
Average MALDI-TOF mass spectra in the mass-to-charge range of lipids acquired after different chemical treatment steps of the lung periphere tissue. Naturally abundant lipids in lung tissues can be partially removed by tissue rinsing in chloroform, xylene, acetone and hexane and can be more completely removed by Carnoy’s washing procedure. Mass spectra were generated in R software (Cardinal).

**Figure 2 mps-02-00044-f002:**
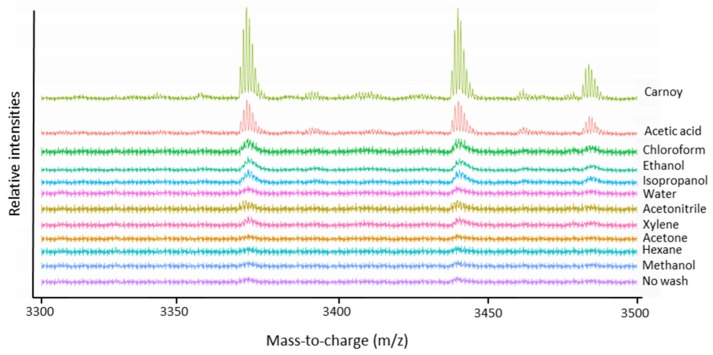
Average MALDI-TOF mass spectra in the mass-to-charge range of three specific endogenous peptides, *m*/*z* 3369.5, *m*/*z* 3440.5 and *m*/*z* 3484.6, acquired after different chemical treatment steps of the lung periphere tissue. Extraction of the three peptides seems to be most efficient using Carnoy’s washing procedure. Mass spectra were generated in R software (Cardinal).

**Figure 3 mps-02-00044-f003:**
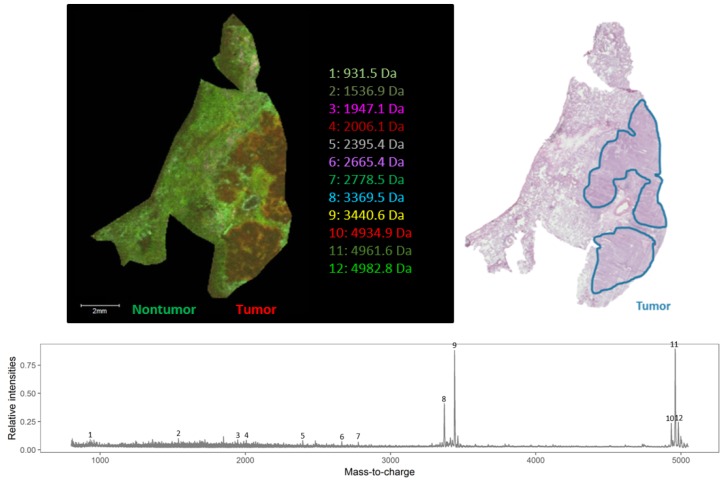
Endogenous peptide profile (*m*/*z* range 800–5000 Da) of human squamous cell carcinoma lung periphere tissue. The tumor and nontumor region can be distinguished based upon the endogenous peptide profile in the lung periphere tissue and these regions are confirmed with the corresponding hematoxylin and eosin (H&E) staining. Peak picking was performed manually and the corresponding peaks were indicated in the mass spectrum. The distribution of the selected individual *m*/*z* values is presented in Figure A1.

**Figure 4 mps-02-00044-f004:**
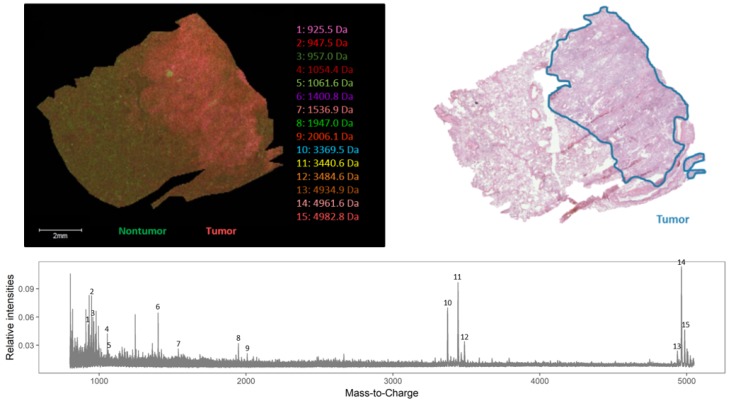
Endogenous peptide profile (*m*/*z* range 800–5000 Da) of human adenocarcinoma lung periphere tissue. The tumor and nontumor region can be distinguished based upon the endogenous peptide profile in the lung periphere tissue and these regions are confirmed with the corresponding H&E staining. Peak picking was performed manually and the corresponding peaks were indicated in the mass spectrum. The distribution of the selected individual *m*/*z* values is presented in Figure A2.

**Figure 5 mps-02-00044-f005:**
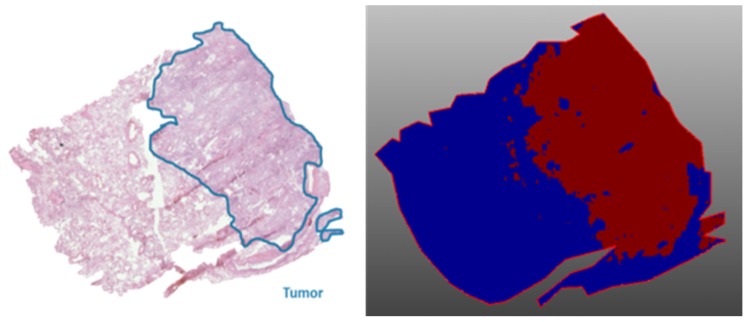
Top-down spatial segmentation test performed in SCiLS lab 2016b on a lung periphere tissue with diagnosis adenocarcinoma (Figure 4). The red region corresponds with the tumor region and the blue region with the corresponding nontumor region; both regions are confirmed by the corresponding H&E staining.

**Figure 6 mps-02-00044-f006:**
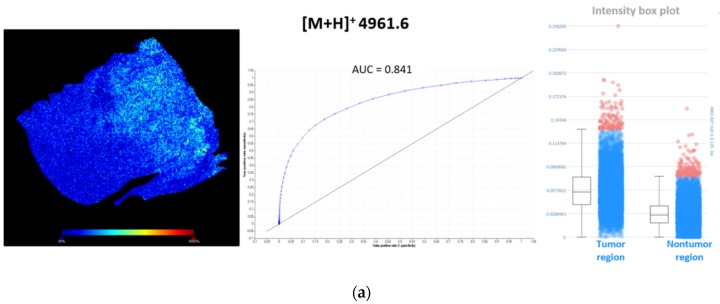
Characteristic *m*/*z* values for the lung tumor region in both periphere tissues. (**a**) ROC analysis, with a discrimination threshold of 0.8, performed on human adenocarcinoma lung periphere tissue characterizes *m*/*z* 4961.6 as a binary classifier to distinguish between lung tumor and nontumor region. This *m*/*z* value showed higher spatial intensities in the lung tumor region compared to nontumor region. A full list of all characteristic *m*/*z* values for lung tumor region in human adenocarcinoma lung periphere tissue is presented in Table A1 (Appendix A); (**b**) ROC analysis, with a discrimination threshold of 0.8, performed on human squamous cell carcinoma lung periphere tissue characterizes *m*/*z* 4934.9 as a binary classifier to distinguish between lung tumor and nontumor region. This *m*/*z* value showed higher spatial intensities in the lung tumor region compared to the nontumor region.

**Figure 7 mps-02-00044-f007:**
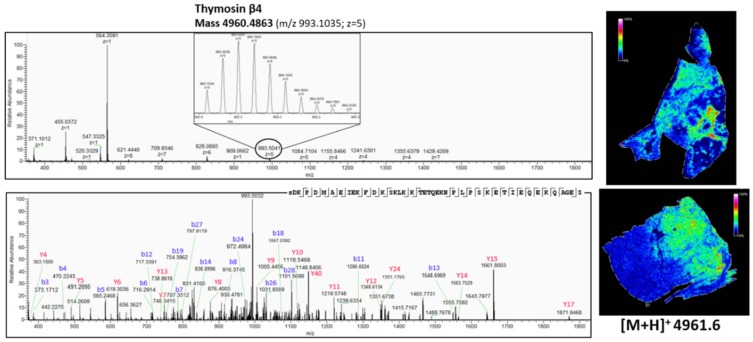
Top-down peptidomics with higher resolution mass spectrometry (Q-Exactive Plus) of MSI target *m*/*z* 4961.6. The precursor (charge 5) is selected in full MS1 scan in the upper panel and is fragmented (HCD) in the lower panel. The corresponding fragmentation spectrum was de novo sequenced with PEAKS software with possible modifications; oxidation, phosphorylation, acetylation, amidation and pyro-glu formed from Q and E. The sequenced peptide was aligned to thymosin β4, which is acetylated at the N-terminus (+42.011 Da). On the right, the corresponding distribution of thymosin β4, obtained with MALDI MSI (Figure 3 and Figure 4), is represented. See also Table A2 in Appendix B for all the matched (deconvoluted) fragment ions.

**Figure 8 mps-02-00044-f008:**
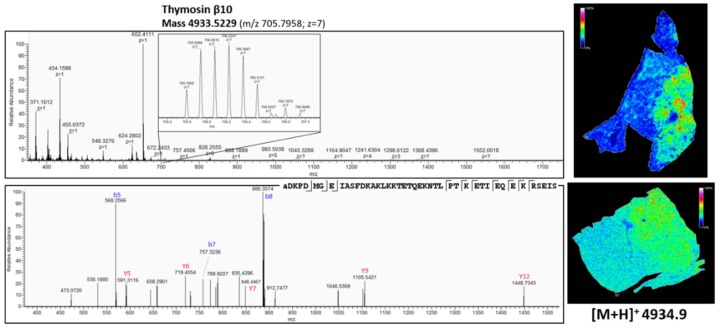
Top-down peptidomics with higher resolution mass spectrometry (Q-Exactive Plus) of MSI target *m*/*z* 4934.9. The precursor (charge 7) is selected in full MS1 scan in the upper panel and is fragmented (HCD) in the lower panel. The corresponding fragmentation spectrum was de novo sequenced with PEAKS software with possible modifications; oxidation, phosphorylation, acetylation, amidation and pyro-glu formed from Q and E. The sequenced peptide was aligned to thymosin β10, which is acetylated at the N-terminus (+42.011 Da). On the right, the corresponding distribution of thymosin β10 obtained with MALDI MSI (Figure 3 and Figure 4) is represented. See also Table A3 in Appendix B for all the matched (deconvoluted) fragment ions.

**Figure 9 mps-02-00044-f009:**
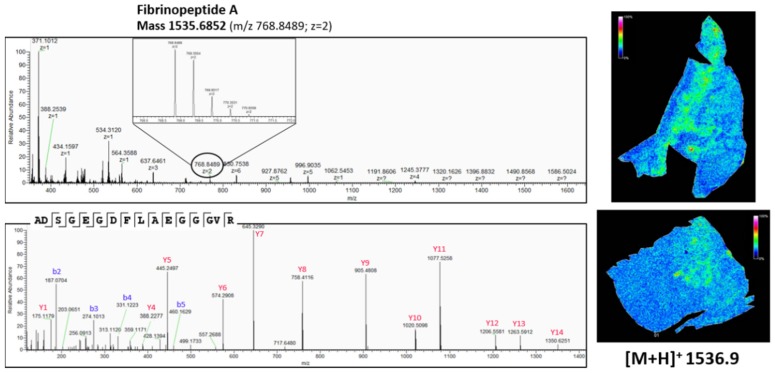
Top-down peptidomics with higher resolution mass spectrometry (Q-Exactive Plus) of MSI target *m*/*z* 1536.9. The precursor (charge 2) is selected in full MS1 scan in the upper panel and is fragmented (HCD) in the lower panel. The corresponding fragmentation spectrum was de novo sequenced with PEAKS software with possible modifications; oxidation, phosphorylation, acetylation, amidation and pyro-glu formed from Q and E. The sequenced peptide was aligned to Fibrinopeptide A, without modifications. On the right, the corresponding distribution of Fibrinopeptide A, obtained with MALDI MSI (Figure 3 and Figure 4), is represented. See also Table A4 in Appendix B for all the matched (deconvoluted) fragment ions.

**Table 1 mps-02-00044-t001:** Different washing procedures tested to obtain good quality mass spectrometry imaging (MSI) images of non-small-cell lung cancer (NSCLC) tissues.

Protocol	Step 1	Step 2	Step 3	Step 4	Ref.
No wash (control)					[29,33,34]
Methanol	30 s 100% MeOH				[35]
Hexane	30 s 100% hexane				[29,31,33]
Acetone	30 s 100% acetone				[33,36]
Xylene	30 s 100% xylene				[21,31,33]
Acetonitrile	30 s 100% AcN				[35]
Water	30 s water (LC-MS graded)				[36]
Isopropanol	30 s 150 mM Ammonium acetate	30 s 100% isopropanol			[30,33,38]
Ethanol	30 s 70% EtOH	30 s 100% EtOH			[21,31,33,34]
Chloroform	30 s 100% chloroform				[21,29,33]
Acetic acid	30 s 70% EtOH	30 s acetic acid buffer ^1^			[21]
Carnoy’s	30 s 70% EtOH	30 s 100% EtOH	90 s Carnoy’s fluid ^2^	30 s 100% EtOH	[37]

^1^ Acetic acid buffer; EtOH:acetic acid:water (90:9:1 v:v:v); ^2^ Carnoy’s fluid; EtOH:chloroform:acetic acid (6:3:1 v:v:v).

**Table 2 mps-02-00044-t002:** Number of detected peptides in lung periphere tissues for every performed wash procedure. Most peptide signals were observed after Carnoy’s washing procedure.

Protocol	# Detected Peptides
No wash (control)	8
Methanol	5
Hexane	11
Acetone	16
Xylene	11
Acetonitrile	9
Water	9
Isopropanol	7
Ethanol	10
Chloroform	10
Acetic acid	18
Carnoy’s	33

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
