# Peer review of "MALDI Mass Spectrometry Imaging Linked with Top-Down Proteomics as a Tool to Study the Non-Small-Cell Lung Cancer Tumor Microenvironment"

_mps, 2019, doi:10.3390/mps2020044_

Round 1

Reviewer 1 Report

mps-484596

MALDI Mass Spectrometry Imaging linked with Top-2 Down Proteomics as a tool to study the Non-Small-3 Cell Lung Cancer Tumor Microenvironment

The article aims in the study of the Non-Small-Cell Lung Cancer (NSCLC) by means of MALDI Mass spectrometry Imaging. The authors have presented a sample preparation method to improve the detection of peptides/small proteins from NSCLC tissue sections.

Even if the article is well written and easy to read, the results provided by the authors are not sufficient to justify the publication of the article. Indeed, the method presented here was already described by previous groups has mentioned in the article in the table 1. The MSI community already knows that for each type of tissue, the sample preparation is the key point and has to be investigated. Moreover, the authors stayed focused on peptides/proteins up to m/z 5000. The reader can understand that it is probably to detect the monoisotopic mass of each ion but the group of Caprioli have already demonstrated that higher mass proteins up to m/z 15000 can be detected and localized using a MALDI-FT-ICR instrument providing high mass accuracy and very high resolution. The detected proteins were then identified by LC-MS/MS using a LTQ-Orbitrap equipped with ETD activation mode.

Recently, it was also demonstrated that MALDI-MSI experiment can be used as a molecular histology tool allowing the determination of regions on interested (ROIs) by means of segmentation as presenting by the authors from NSCLC tissue sections. Proteins can be therefore extracted from discrete location from the determined ROIs leading to the identification and quantification of more than 2000 proteins.

Considering that the development presenting here including the washing steps cannot provide gain in benefit in the field of the MSI, the article cannot be accepted as publication. Moreover, the m/z range measured is too limited for a real top-top approach and the number of detected ions is too poor to really reflect a pathological state.

The authors should use their optimization to gain into insight in the biological part. The MALDI-MSI could therefore be used as a molecular histology tool to detect ROIs and to visualize the response to a specific treatment. This can also be obtained by MALDI-MSI of lipids which required less sample preparation. From the determined regions obtain by segmentation, proteins can be extracted and identify allowing a large scale proteomics analysis.

Author Response

General remarks:

We fear the reviewer has in part missed the point of our study. Our aim clearly was to visualize the distribution of small (secretory) proteins and/or peptides that play a role in the regulation of the immune response in NSCLC. The aim, in the end,  is to use MSI to 1) obtain better insight in the immune response to the tumor in relation to immunotherapy and 2) set-up and evaluate MSI as a tool for discovery of therapy response biomarkers. As such the aim is to 1) set-up a method to image peptides in lungs (we are at the moment not interested in lipids although we know they do generate nice images) 2) pick a technique that can be used to generated good quality images of a “large” number of samples for the biobank in a reasonable amount of time (hence the choice for MALDI-TOF over FTICR).

As pointed out by the reviewer “for each type of tissue, the sample preparation is the key point and has to be investigated”;  To this day, very little MSI data on lung exists, especially for intact peptides. The aim of this study is to set-up and evaluate methods for imaging of endogenous peptides in lung taking into account the requirements for future studies. The determination of ROIs (remark No 3) is for us a tool to evaluate the sample preparation (extend of lateral diffusion, sensitivity, etc..) rather than a method to distinguish tumor from normal tissue. Other and better methods than MSI exist to do this.

Specific remarks:

1. Even if the article is well written and easy to read, the results provided by the authors are not sufficient to justify the publication of the article. Indeed, the method presented here was already described by previous groups has mentioned in the article in the table 1. The MSI community already knows that for each type of tissue, the sample preparation is the key point and has to be investigated.

As explained in general remarks, little data exist on MSI of peptides in lung. On page 16, lines 477 – 479, we included a paragraph to emphasize that the first aim of this study was to evaluate endogenous peptide imaging in human lung tissue, on which very little data exist to this day. We think this makes our study quite relevant indeed!

2. Moreover, the authors stayed focused on peptides/proteins up to m/z 5000. The reader can understand that it is probably to detect the monoisotopic mass of each ion but the group of Caprioli have already demonstrated that higher mass proteins up to m/z 15000 can be detected and localized using a MALDI-FT-ICR instrument providing high mass accuracy and very high resolution. The detected proteins were then identified by LC-MS/MS using a LTQ-Orbitrap equipped with ETD activation mode.

This is an interesting remark and we agree with the reviewer that we could have discussed this in more detail.

In general, assuming that the reviewer refers to Spraggins et al, 2015. We are well aware of the fact that the added mass accuracy can help linking MSI data to LC-MS/MS top-down data. Actually, we made similar observations in our own work that predates the study of Caprioli et al. (Minerva L, et al; Anal Chem. 2011 Oct 15;83(20):7682-91). However, both the study of Caprioli’s group and ours was on endocrine tissue (brain and pancreas) in which large quantities of natural peptides are to be expected and high resolution is needed to separate the different peptide species. We are well aware of this since in one of our recent papers, we identified close to 1000 peptides in mouse brain using top-down peptidomics (Budamgunta et al 2018), so one might argue that resolution of FTICR is not sufficient to detect all natural peptides in brain. However, this is not necessarily the case in lung which does not have an obvious endocrine function and lower numbers of peptides are to be expected, in which case MALDI-TOF specifications are adequate and the speed of MALDI-TOF is a requisite to analyse large numbers of samples. A paragraph was added in the discussion (page 16, lines 495 - 509) to discuss the work of Caprioli and compare it to our study and discuss differences between FTICR and TOF.

3. Recently, it was also demonstrated that MALDI-MSI experiment can be used as a molecular histology tool allowing the determination of regions on interested (ROIs) by means of segmentation as presenting by the authors from NSCLC tissue sections. Proteins can be therefore extracted from discrete location from the determined ROIs leading to the identification and quantification of more than 2000 proteins.

We assume the reviewer is referring to in situ digest analysis as shown extensively by the group of Fournier and Salzet. Although of great relevance to the field of MALDI imaging, as explained earlier, we are interested specifically in endogenous peptides/small regulatory proteins, not in the entire proteome.

4. Considering that the development presenting here including the washing steps cannot provide gain in benefit in the field of the MSI, the article cannot be accepted as publication. Moreover, the m/z range measured is too limited for a real top-top approach and the number of detected ions is too poor to really reflect a pathological state.

We do believe it is of relevance since little data does exist on lung and as pointed out by the reviewer, the sample treatment is of key importance. We indeed observe largely “small” peptides. Very little larger peptides could be observed, which could be due to limitations of the sample preparation protocol but also, and probably more so, due to the absence or low concentrations and LOD of larger peptides in lung. It is definitely not caused by limitation of the technique as proteins up to 30 kDa are routinely detected in other tissues with MALDI-TOF MSI. As discussed before, very little data exists on human lung so it is difficult to compare.

5. The authors should use their optimization to gain into insight in the biological part. The MALDI-MSI could therefore be used as a molecular histology tool to detect ROIs and to visualize the response to a specific treatment. This can also be obtained by MALDI-MSI of lipids which required less sample preparation. From the determined regions obtain by segmentation, proteins can be extracted and identify allowing a large scale proteomics analysis.

As explained in the general remarks we are interested mainly in peptides, not in lipids or (larger) proteins.

Reviewer 2 Report

The manuscript by Eline Berghmans et.al describes a MALDI mass spectrometry imaging-based peptidomic approach to characterize endogen peptides in Non-Small-Cell Lung Cancer (NSCLC) tissues. Authors tested different sample preparation procedures and found the Carnoy’s treatment as the best procedure to enhance peptide extraction without causing signal delocalization. A peptide signature discriminating non-tumor from tumor regions has been shown and some of discriminant peptides were identified using a top-down peptidomics approach. With their findings the authors concluded that combining mass spectrometry imaging and top-down peptidomics is a potential approach to fully understand the molecular profile of the NSCLC tumor microenvironment.

The study has been clearly written, and the results interesting. Moreover, I recommend publication pending the authors take into consideration the following comments:

Page 2 lines 78-79: in the sentence “(MALDI) mass spectrometry imaging (MSI) is a powerful tool to produce reliable images of the spatial distribution from a broad variety of biomolecules (e.g. peptides, proteins, lipids, nucleic acids and metabolites)” please include glycans as other biomolecules analyzed by MALDI MSI. Include also a reference such as PMID: 28110657, PMID: 27373711.

Page 3 lines 99-100: The authors stated “MSI data of lung (cancerous) tissues is currently limited to drug distribution [28,29] or lipidomic profiling [30].” This is not true. There are several articles showing MSI-based proteomic data of tumor lung such as “PMID: 27939606, PMID: 27473201, PMID: 27228175, PMID: 18712763 among others. Please, rewrite this sentence including a “proteomic profile”, as another kind of MSI data currently existing, together with an appropriate reference.

Page 4 lines 154-155, subsection “2.1 Materials”, how many tissues were used in the study, and how many adenocarcinoma and squamous cell carcinomas were measured? Please specify.

Page 4 line 180, “…using a SunCollect pneumatic sprayer…”. Please include more details in regards of parameter used for the matrix spraying with the SunCollect such as flow rate, air pressure, drying time if any.

Subsection “2.4. H&E staining”: In the first sentence it is stated that one section was stained, while in the last sentence the authors stated “The tissue sections were in this way re-evaluated by the pathologist…”. How many sections were then stained, one or more? Also, what is the ratio of the reagents in the AFA solution? Please specify.

Subsection “2.5. Peptide and intact small protein identification”: How many tissue slices were collected from the same tissue, and how many individual tissues were used for the identification analysis? Also, how long was the sample kept in the peptide extraction solvent? For how long was the sample sonicated? Which relative centrifugal force (rcf) as well as temperature and running time were used for the centrifugation? Please include these parameters.

In the subsection “3.1. Comparison of different chemical treatment steps of lung cancer tissues”, page 6 lines 273-275, the authors stated “Probably these solvent wash procedures are too harsh and wash out the peptides…” However, also in the “No wash” test there were no peptides detected as showed in Figure 2, how can the author explain this? “…or the removal of the lipids is not efficient enough, which can lead to ion suppression effects due to high ion intensity of the lipids”. This reviewer agreed with the authors if the mass range considered was the typical lipids range (m/z 800-1000) as it is shown in Figure 1. There should not be ion suppression effect due to lipids in the high mass range 3300-3500, where usually lipids are not detected. Please discuss/clarify this statement.

“…. is apparent that Carnoy’s washing procedure leads to the most complete lipid removal and thereby yielding the best endogenous peptide signal intensity.” How many peptides were detected with the Carnoy’s washing procedure, and what was the total number of peptides detected in each washing procedure tested and eventually in the “No wash” control?

Please include the scale bar for each MS image in Figures A1 and A2.

Subsection “3.2. Evaluation of peptide delocalization: cancerous from noncancerous lung tissue”: Assessing that there is no delocalization based on the fact that specific peaks distribute in the tumor or non-tumor regions is arguable. How can the authors prove that the ions detected in a specific region, for example the tumor region, are specifically related to the tumor? Were all these peaks identified and so their specific tissue distribution explained based on comprehensive examinations? How the spectra profile looks like around the edges of the tissue? Was any peptide signal observed off tissue? This reviewer would like to see a comparison between an average spectrum from outside and inside of the tissue boundary. This comparison would better show the non-delocalization phenomenon.

Page 10, lines 324-327: Were the specific peptides related to the different tumors (adenocarcinoma and squamous cell carcinoma) detected in the all ten tissues measured?

Please provide false discovery rate as well as score parameters used for the identification of the peptides.

Page 10, lines 346-348. “ROC analysis was performed with a discrimination threshold value of 0.8 to obtain a peak list of discriminative m/z values.” Please provide a table showing the peak list of discriminative m/z values with their corresponding ROC.

Author Response

The manuscript by Eline Berghmans et.al describes a MALDI mass spectrometry imaging-based peptidomic approach to characterize endogen peptides in Non-Small-Cell Lung Cancer (NSCLC) tissues. Authors tested different sample preparation procedures and found the Carnoy’s treatment as the best procedure to enhance peptide extraction without causing signal delocalization. A peptide signature discriminating non-tumor from tumor regions has been shown and some of discriminant peptides were identified using a top-down peptidomics approach. With their findings the authors concluded that combining mass spectrometry imaging and top-down peptidomics is a potential approach to fully understand the molecular profile of the NSCLC tumor microenvironment.

The study has been clearly written, and the results interesting. Moreover, I recommend publication pending the authors take into consideration the following comments:

1. Page 2 lines 78-79: in the sentence “(MALDI) mass spectrometry imaging (MSI) is a powerful tool to produce reliable images of the spatial distribution from a broad variety of biomolecules (e.g. peptides, proteins, lipids, nucleic acids and metabolites)” please include glycans as other biomolecules analyzed by MALDI MSI. Include also a reference such as PMID: 28110657, PMID: 27373711.

Analysis of glycans by MALDI MSI is included in this sentence (referenced by Heijs, B. ; Holst, S. ; Briaire-De Bruijn, I.H. ; Van Pelt, G.W. ; De Ru, A.H. ; Van Veelen, P.A. ; Drake, R.R. ; Mehta, A.S. ; Mesker, W.E. ; Tollenaar, R.A. ; Bovée, J.V.M.G. ; Wuhrer, M. & McDonnell, L.A. Multimodal Mass Spectrometry Imaging of N-Glycans and Proteins from the Same Tissue Section. Anal. Chem. 2016, 88, 7745–7753). 

2. Page 3 lines 99-100: The authors stated “MSI data of lung (cancerous) tissues is currently limited to drug distribution [28,29] or lipidomic profiling [30].” This is not true. There are several articles showing MSI-based proteomic data of tumor lung such as “PMID: 27939606, PMID: 27473201, PMID: 27228175, PMID: 18712763 among others. Please, rewrite this sentence including a “proteomic profile”, as another kind of MSI data currently existing, together with an appropriate reference.

Indeed, we agree with the reviewer that it is appropriate to discuss this data as well. This sentence is rewritten with the addition of proteomic profiling on FFPE lung tissues (reference proteomic profiling (on FFPE tissues)’ is also added as MSI data on lung tissues with reference [32] (Kriegsmann, M. ; Casadonte, R. ; Kriegsmann, J. ; Dienemann, H. ; Schirmacher, P. ; Kobarg, J.H. ; Schwamborn, K. ; Stenzinger, A. ; Warth, A. & Weichert, W. Reliable entity subtyping in Non-small cell Lung Cancer by MALDI Imaging Mass Spectrometry on Formalin-fixed Paraffin-embedded Tissue Specimens. Mol. Cell. Proteomics 2016, 15, 3081–3089)

3. Page 4 lines 154-155, subsection “2.1 Materials”, how many tissues were used in the study, and how many adenocarcinoma and squamous cell carcinomas were measured? Please specify.

More detailed information on the used tissues, from which data was displayed throughout the paper, were included in the section ‘2.1 Materials’. Shown data were derived from two lung cancer patients diagnosed as squamous cell carcinoma and from one patient diagnosed as adenocarcinoma.

On page 11, line 334, it was stated that up to ten different lung periphere tissues were measured (not all data shown). From these ten lung periphere tissues, diagnosis information was added: 4 adenocarcinoma and 6 squamous cell carcinomas.

4. Page 4 line 180, “…using a SunCollect pneumatic sprayer…”. Please include more details in regards of parameter used for the matrix spraying with the SunCollect such as flow rate, air pressure, drying time if any.

 In the section ‘2.1 Materials’ the following, more detailed information was added: ‘Matrix layers were applied with an air pressure of 2 bar and with a flow rate of 10 µl/minute for the first layer, followed by 20 µl/min for the second layer, 30 µl/min for the third and from then on 50 µl/min, without any drying time between the matrix layers’. The height of the sprayer (35 mm) was mentioned in section ‘3.2 Evaluation of peptide delocalization: cancerous from noncancerous lung tissue’ on page 11 – line 345.

5. Subsection “2.4. H&E staining”: In the first sentence it is stated that one section was stained, while in the last sentence the authors stated “The tissue sections were in this way re-evaluated by the pathologist…”. How many sections were then stained, one or more? Also, what is the ratio of the reagents in the AFA solution? Please specify.

Each tissue was H&E stained after the MSI experiment. The AFA solution consists of 70%EtOH;5% acetic acid:5% formalin v:v:v). This information is added in subsection 2.4. H&E staining.   

6. Subsection “2.5. Peptide and intact small protein identification”: How many tissue slices were collected from the same tissue, and how many individual tissues were used for the identification analysis? Also, how long was the sample kept in the peptide extraction solvent? For how long was the sample sonicated? Which relative centrifugal force (rcf) as well as temperature and running time were used for the centrifugation? Please include these parameters.

These parameters are included and are the following: 10 individual tissue slices were used for identification analysis, the sample is shaked on ice for 30 minutes, sonicated twice for 15 seconds and a centrifugation step of 15 minutes at 14000 rcf by 4°C.

7. In the subsection “3.1. Comparison of different chemical treatment steps of lung cancer tissues”, page 6 lines 273-275, the authors stated “Probably these solvent wash procedures are too harsh and wash out the peptides…” However, also in the “No wash” test there were no peptides detected as showed in Figure 2, how can the author explain this? “…or the removal of the lipids is not efficient enough, which can lead to ion suppression effects due to high ion intensity of the lipids”. This reviewer agreed with the authors if the mass range considered was the typical lipids range (m/z 800-1000) as it is shown in Figure 1. There should not be ion suppression effect due to lipids in the high mass range 3300-3500, where usually lipids are not detected. Please discuss/clarify this statement.

Even the peptides in the mass range 3300 – 3500 can be harder to detect when lipids are still present in the tissue; since ion suppression occurs in the source, even if the lipids are excluded from the observed mass range, their presence alone will suppress intensity of the peptide signal.

In the ‘4. Discussion’ section, we added an extra explanation why we are more likely to detect and measure peptides in the tissues, when the lipids were removed from the sample. Page 16, lines 486 – 487 are stating now: ‘In this way, the phospholipids, who build up the most of the cell membrane, were removed and endogenous (mostly secretory) peptides could be displayed to the tissue surface. 

8. “…. is apparent that Carnoy’s washing procedure leads to the most complete lipid removal and thereby yielding the best endogenous peptide signal intensity.” How many peptides were detected with the Carnoy’s washing procedure, and what was the total number of peptides detected in each washing procedure tested and eventually in the “No wash” control?

In the rebuttal, an extra table (Table 2) is added that displays the total number of peptides detected in each sample after the 12 different sample preparation steps. Also here it is clear that the Carnoy’s washing procedure leads to the detection of the most peptides. 

This is also included in the main text on page 6, lines 273-278: ‘The evaluation of the endogenous peptide profile included amount of detected peptides (see Table 2) and peptide extraction efficiency. This last part was limited to the m/z range 3300-3500 in order to compare the extraction efficiency of three intense peptide signals m/z 3369.5, m/z 3440.6 and m/z 3484.6 that occurred in all samples. As shown in Figure 2, these three peptides were best visible using Carnoy’s washing procedure and acetic acid treatment. Of these two procedures, both most peptide signals (Table 2) as highest signal intensity of the peptide peaks were observed using Carnoy’s washing procedure.’

9. Please include the scale bar for each MS image in Figures A1 and A2.

Figures A1 and A2 were updated in the rebuttal with an intensity scale bar for each peptide signal.

10. Subsection “3.2. Evaluation of peptide delocalization: cancerous from noncancerous lung tissue”: Assessing that there is no delocalization based on the fact that specific peaks distribute in the tumor or non-tumor regions is arguable. How can the authors prove that the ions detected in a specific region, for example the tumor region, are specifically related to the tumor? Were all these peaks identified and so their specific tissue distribution explained based on comprehensive examinations? How the spectra profile looks like around the edges of the tissue? Was any peptide signal observed off tissue? This reviewer would like to see a comparison between an average spectrum from outside and inside of the tissue boundary. This comparison would better show the non-delocalization phenomenon.

Please find the attached mass spectra, one mass spectrum corresponding for peptides measured on the tissue itself, the other spectrum for off-tissue region. This is another evidence that there is no delocalization of the peptides, even as the segmentation test performed in the paper. This figure is included in the rebuttal in appendix A (Figure A3). Each individual peptidomic profile of a NSCLC patient is different, but some common peaks are observed throughout different NSCLC tissues, each representing or the tumor region, or the nontumor region (both regions are for every tissue confirmed with a H&E staining).

11. Page 10, lines 324-327: Were the specific peptides related to the different tumors (adenocarcinoma and squamous cell carcinoma) detected in the all ten tissues measured?

Each individual peptidomic profile of a NSCLC patient is different, but some common peaks are observed throughout different NSCLC tissues, each representing or the tumor region, or the nontumor region (both regions are for every tissue confirmed with a H&E staining). To make this more clear, we added ‘individual’ peptide profile on page 11, line 337.

Also, expression information about the three identified peptides is added for the all ten measured tissues. Both thymosin β10 and thymosin β4 were expressed in all ten measured tissues. This is added on page 15, lines 440-441. Page 15, lines 445-447 contains information about fibrinopeptide A expression levels: ‘From all the ten measured NSCLC patient tissues, fibrinopeptide A was expressed in three different patients with a normal intensity and in two patient samples with a low intensity. In the remaining five NSCLC samples no fibrinopeptide A could be detected.’

As all the NSCLC samples were used for optimizing our methods, we do not have known treatment/prognosis information of each of them. Therefore, we cannot further conclude therapy response based on the expression levels of these identified peptides, but literature provides an appealing biological explanation for the observed expression profiles, that illustrates that MSI combined with top-down can be used to elucidate new response patterns for cancer treatment.  

12. Please provide false discovery rate as well as score parameters used for the identification of the peptides.

These parameters are included on page 6, lines 250-251: ‘with 1% false discovery rate. Parent mass error tolerance was set at 10 ppm and precursor mass error tolerance at 0.02 Da’, even as the -10lgP values corresponding for each identified peptide were mentioned in tables B1, B2 and B3 in appendix B.

13. Page 10, lines 346-348. “ROC analysis was performed with a discrimination threshold value of 0.8 to obtain a peak list of discriminative m/z values.” Please provide a table showing the peak list of discriminative m/z values with their corresponding ROC.

In appendix A, Table A1 is added which provides a peak list of all discriminative m/z values with their corresponding AUC. In the main text is referred to this table A1 on page 11, lines 360 – 365: ‘, with their corresponding ‘area under the ROC curve (AUC)’ scores, which respresent the accuracy of the test; the closer to 1, the more perfect. A full list of all characteristic m/z values for lung tumor region in human adenocarcinoma lung periphere tissue is presented in table A1 (Appendix A). For the lung periphere tissue with human squamous cell carcinoma diagnosis, only m/z value 4934.9 was found to be discriminative between tumor and nontumor region.’ Also in the Figure’s text (of Figure 6), the table that contains all characteristic m/z values is mentioned: ‘A full list of all characteristic m/z values for lung tumor region in human adenocarcinoma lung periphere tissue is presented in table A1 (Appendix A)’.

Reviewer 3 Report

This paper demonstrates different distribution of some peptides in tumor and non-tumor region by linking localization of the peptides in mass spectrometry imaging (MSI) and their identification of the molecules based on top-down peptidomics using LC-MS/MS.  Although the biological meanings of the difference in distribution of thymosin beta4, thymosin beta10, fibrinopeptide A are not investigated at all, the combination of MSI and top-down approaches could be useful for discovery of novel biomarkers in cancer.

Author Response

This paper demonstrates different distribution of some peptides in tumor and non-tumor region by linking localization of the peptides in mass spectrometry imaging (MSI) and their identification of the molecules based on top-down peptidomics using LC-MS/MS.  Although the biological meanings of the difference in distribution of thymosin beta4, thymosin beta10, fibrinopeptide A are not investigated at all, the combination of MSI and top-down approaches could be useful for discovery of novel biomarkers in cancer.

Biological meanings of the difference in distribution were indeed not investigated, as all measured NSCLC tissues were test samples for optimizing our methods, of which treatment/prognosis is not (yet) known. In this context, we cannot further conclude therapy response/biological meanings of the expression levels of these identified peptides, but literature provides an appealing biological explanation for the observed expression profiles, that illustrates that MSI combined with top-down can be used to elucidate new response patterns for cancer treatment. We added extra information about fibrinopeptide A expression levels in all ten measured NSCLC tissues (Page 15, lines 445-447): ‘From all the ten measured NSCLC patient tissues, fibrinopeptide A was expressed in three different patients with a normal intensity and in two patient samples with a low intensity. In the remaining five NSCLC samples no fibrinopeptide A could be detected’. This difference in expression profile of fibrinopeptide A was supported in literature, as mentioned before.

Round 2

Reviewer 1 Report

The responses provided by the authors are sufficient to accept the article for publication.

Reviewer 2 Report

The revised manuscript has been improved. The authors addressed satisfactorily all concerns I had. I find this manuscript acceptable for publication.